# Radar-Based Heart Cardiac Activity Measurements: A Review

**DOI:** 10.3390/s24237654

**Published:** 2024-11-29

**Authors:** Alvaro Frazao, Pedro Pinho, Daniel Albuquerque

**Affiliations:** 1Department of Electronics Telecommunications and Informatics (DETI), Instituto de Telecomunicações, Universidade de Aveiro, 3810-193 Aveiro, Portugal; ptpinho@ua.pt; 2Águeda School of Technology and Management (ESTGA), Instituto de Telecomunicações, Universidade de Aveiro, 3750-127 Águeda, Portugal; dfa@ua.pt

**Keywords:** Doppler radar, HRV, remote vital signs, CW, FMCW, UWB

## Abstract

In recent years, with the increased interest in smart home technology and the increased need to remotely monitor patients due to the pandemic, demand for contactless systems for vital sign measurements has also been on the rise. One of these kinds of systems are Doppler radar systems. Their design is composed of several choices that could possibly have a significant impact on their overall performance, more specifically those focused on the measurement of cardiac activity. This review, conducted using works obtained from relevant scientific databases, aims to understand the impact of these design choices on the performance of systems measuring either heart rate (HR) or heart rate variability (HRV). To that end, an analysis of the performance based on hardware architecture, carrier frequency, and measurement distance was conducted for works focusing on both of the aforementioned cardiac parameters, and signal processing trends were discussed. What was found was that the system architecture and signal processing algorithms had the most impact on the performance, with FMCW being the best performing architecture, whereas factors like carrier frequency did not have an impact.This means that newer systems can focus on cheaper, lower-frequency systems without any performance degradation, which will make research easier.

## 1. Introduction

Over the last few years, the need to remotely monitor the vital signs of people has been on the rise. Whether this is due to the possible appearance of another highly contagious disease, or the increasing interest in smart home technology to assist in providing healthcare for the elderly, this technology presents plenty of relevant use cases that justify giving attention to the topic. These systems are usually designed to measure certain physiological parameters, for example, respiratory and cardiac rates in a person or the variability of their heart rate. Traditionally, these measurements are performed either by direct observation of a patient, as is the case of respiratory rate measurements, or by direct contact through sensors placed on the patient’s body for cardiac measurements. Despite providing the best results, these methods are sometimes not suitable due to requiring direct contact and or proximity to the patient, being impractical, being unusable (e.g., they cannot be used for infants and people with skin conditions or burns), and just being generally uncomfortable for the patient; this can also negatively affect the results. Remote measurement systems tackle these issues, since they are able to obtain results without causing any hindrance to the patients in question. Over the years, several different kinds of remote sensing systems have been developed, such as ones based on thermal or optical cameras that extract data from the changes in skin temperature or color [1,2], Doppler radar systems [3], ultrasonic systems [4], and systems that leverage the channel state information in Wi-Fi networks to extract physiological signals [5]. However, camera- and Wi-Fi-based systems present problems relative to privacy and light/temperature interference which make radar systems the better candidate for remote vital sign sensing. Radar still presents its own set of challenges, but these can be more easily dealt with by employing the appropriate hardware and software implementations of the system, usually defined on a case-by-case basis. Due to this myriad of not only use cases but also system implementations, research on this topic is very widespread and, as such, several reviews have already been carried out on this topic; they are detailed below.

Obadi et al. [6] compared the implementations of these types of radar systems both from a hardware and signal processing perspective, concluding with an analysis of the existing implementations, which was performed using Field Programmable Gate Arrays (FPGAs); this required the consideration of points that are usually disregarded in the literature, such as power consumption and the computational complexity of the used algorithms. Zhang et al. [7] carried out a review on the state of the art of implementations based on the target objective of the given system, such as vital sign monitoring, activity detection, or noise mitigation. An analysis of the trends in radar hardware architecture and signal processing chains was also performed, the latter differentiating between each class of algorithms (spectrum and periodicity, among others), in order to assist in understanding which situations each architecture and algorithm are best suited to. A review of existing public datasets was performed, and lastly, a discussion about the current challenges that radar-based cardiac monitoring faces, as well possible future work to solve them, was carried out. Wu et al. [8] listed the main challenges that affect the several parts of the signal processing chain, such as data collection, feature extraction, and parameter estimates; they discussed possible solutions for these issues and concluded with a review of the deep learning approaches used in vital sign monitoring. Liebetruth et al. [9] reviewed the current state of the art, analyzing testing conditions, the reference measurement methods used, and the error metrics used in the literature, for the three main hardware architectures (continuous wave (CW), frequency-modulated continuous wave (FMCW)/linear frequency-modulated continuous wave (LFMCW), and ultra wideband (UWB)). In summary, most recent reviews focus on analyzing and comparing the several types of system implementations, from the architecture to the signal processing chain, but they mostly focus on their use in monitoring respiratory and cardiac rates. To the best of the authors’ knowledge, no review has been conducted that analyzes implementations directed specifically towards the measurement of heart rate variability (HRV), nor has one been carried out that assesses the impact that the several parts of the system have on its performance. As such, in terms of the contribution made by this work, this review aims to fill that gap by providing an analysis of the current trends in HRV and HR monitoring and comparing them in order to understand what design choices have the biggest impact on the performance of the system and whether or not HRV monitoring requires a different approach in order to obtain quality measurements.

The rest of the work is organized as follows: Section 2 provides a short overview of the different radar hardware architectures that are more commonly used, as well as a brief definition of HRV; Section 3 contains the listing of the reviewed literature, alongside an explanation for their inclusion; in Section 4, the results of the review are discussed; in Section 5, conclusions are taken from the discussion, and future work is proposed for the challenges presented.

## 2. Background Overview

Radar refers to systems that, through sending and receiving electromagnetic waves into the environment, are able to detect and sense the range of objects therein. Although these systems were initially used to help prevent ship collisions and for aircraft detection, in more recent years, the use of radars has expanded to applications such as weather analysis and measuring vital signs. The measurement of vital signs using radar technology leverages the displacement of the chest wall to perform its measurements which, according to the Doppler effect, causes a shift in the frequency of the reflected radar signal that can also be interpreted as a phase shift [10]. It is by analyzing the phase shift in the signal that parameters like the respiratory and cardiac rate are extracted.

In the remainder of this section, an overview of the different types of radar architectures will be presented, followed by a general overview of the physiological components relevant to these measurements.

### 2.1. Radar Architectures

Although similar in principle, as illustrated in Figure 1, the several architectures are distinguished by the way with which they emit and receive the signals that are sent out into the environment. Based on what is said in [10], CW radars transmit a signal with a constant frequency, FMCW radars transmit a signal where the frequency is modulated over time according to a specific modulation type (linear, stepped), and UWB radars function by emitting pulses with a short duration and a wide-frequency bandwidth. The information regarding the vital signs of the subjects is extracted either from the phase of the reflected signal in the case of CW and FMCW systems, or from the propagation delay of the signal in the case of UWB systems.

Each use case has its unique requirements and as such, each architecture can be used properly if chosen for the right scenario; CW radars are simple in design, but since there is no modulation, it is impossible to measure the range of objects from the reflected signals [10,11]. This means that, if there are multiple targets in range, the signal will suffer distortion, and extra signal processing is required to extract the results. FMCW radars are capable of measuring this range, allowing for target separation and for the mitigation of noise due to reflections from surrounding objects. UWB radars allow for through-wall sensing and monitoring over long distances due to the wide frequency spectrum in its pulses. The downside is that this increases the hardware complexity of the system. However, this has gradually become of a less limiting factor due to the increased availability of hardware platforms that make it easier to implement these architectures [7].

Another important aspect of the design of these systems is the carrier frequency. This is because it not only affects the design of the antennas that are used with the radar, but it also affects the sensitivity to the phase changes and noise of the radar signal. As the carrier frequency increases, the phase changes resulting from smaller movements like the ones caused by cardiac activity also increase. However, this brings the downside of increased sensitivity to noise, meaning that while the phase change from cardiac movements is more noticeable in the reflected signal, so is the impact of random body movements by the subject. The distortion caused by these movements becomes greater as a consequence of this and requires more extensive signal processing to properly mitigate the resulting errors.

### 2.2. Heart Rate Variability

Heart rate variability (HRV) is defined as the variability of the intervals between consecutive heart beats, as seen in Figure 2 and as stated in [12]; this is caused by the autonomic nervous system and the interactions between the brain and the heart. Because of this, it can be used to infer something about a person’s physiological and psychological state. There are a myriad of parameters that allow for the assessment of HRV, which are split between two domains: the frequency and time domain. For time domain measurements, the basis of all parameters is the beat-to-beat interval (BBI), which is then used to calculate parameters such as the standard deviation of N-N intervals (SDNN), the root mean square of successive interval differences (RMSSDs), and the percentage of intervals that differ by more than 50 ms (pNN50). The frequency domain parameters are obtained by analyzing the spectrum of the signal, usually an electrocardiogram, within specific frequency bands. However, the frequency domain parameters are usually not calculated. This is because when using radars, the only phenomenon that is measured for the heart is the vibration it causes on the chest wall, as opposed to an ECG, where the measured signal contains information related to every single step of cardiac activity. As such, the recovered signals will not have enough information to properly calculate these parameters. With this in mind, these parameters will be disregarded in the reviewed works (whether they are measured or not).

## 3. Materials and Methods

In this section, the listing of the reviewed works is carried out. Since the review focused on the system design for both HR and HRV measurement systems, these are listed separately. Only information on the factors relevant to this review is listed, since others like measurement time and number of subjects under testing do not necessarily impact the performance of the system. These would serve only to assess the reliability of the results for any given work. However, since no information regarding the performance can be extracted from these factors, these will not be considered during the analysis for this review.

The works that were reviewed were obtained via Scopus (TITLE-ABS-KEY (“vital signs”) AND TITLE-ABS-KEY (cardiac OR “heart rate variability” OR hrv OR bbi OR “Beat to Beat Intervals” OR “Heart Rate” OR hr) AND TITLE-ABS-KEY (radar OR “Doppler radar” OR “non-contact monitoring”) AND PUBYEAR > 2017 AND PUBYEAR < 2025) and via IEEExplore (“All Metadata”: “vital signs” AND (“All Metadata”: cardiac OR “All Metadata”: “Heart Rate Variability” OR “All Metadata”: hrv OR “All Metadata”: bbi OR “All Metadata”: “beat to beat intervals” OR “All Metadata”: “heart rate” OR “All Metadata”: hr) AND (“All Metadata”: radar OR “All Metadata”: “doppler radar” OR “All Metadata”: “non-contact monitoring”)). The characteristics that were analyzed for similarities were the system architecture, carrier frequency, and measurement distance. While the first two are the most important, they are implementation-specific. The distance was also catalogued and analyzed in order to not only try and assess the impact it has on the measurements but also to assess the conditions in which the measurements are usually done, since, for a technology with such a wide range of possible uses, it is necessary to test it in an equal amount of varied scenarios. The number of subjects is also listed to provide some insight into how extensively each system is tested. With regard to the application scenarios in the reviewed works, most perform measurements in a laboratory environment with the subjects sitting upright and static during the experiments. As such, the scenarios are not specified in the tables. The results of this review are listed in Table 1 and Table 2, and the yearly distribution of the reviewed works is displayed in Figure 3 where, based on the large number of works published in recent years, it is evident that cardiac activity measurement with radar is a relevant research topic that is attracting a lot of attention.

## 4. Results

In this section, the results of the analysis of the reviewed works are presented. Firstly, the works are split into two separate categories depending on whether they measure HRV or not. Within each category, the architectures and carrier frequencies are analysed and compared in order to establish any possible trends in system development and also to try to find a correlation between these parameters and the quality of the measurements made in their respective works, which is one of the main goals of this work. During the analysis of the works listed above, there was a noticeable lack of consistency in terms of the error metrics that researchers used to present their results, which makes a broader comparison of all works more difficult and makes it impossible to compare certain works within a category or subcategory. As such, when performing any sort of quantitative analysis of the reported error metrics, a criterion had to be established in order to be able to compare these results beyond a general qualitative assessment. The chosen criterion was to determine the most used type of error metric within each category and only consider works that reported this metric when performing comparisons. In the case of the two aforementioned categories, the metrics used were the mean absolute error (MAE) for HR-focused systems and the mean relative error (MRE) for HRV-focused systems. In order to increase the amount of data that could be analysed in the first category, the MRE of some works was converted into a corresponding MAE. However, due to some works not reporting a mean HR, a reference mean HR of 70 bpm was used to convert between the two metrics. For HRV focused works this was not possible to do since the second most reported error metric is the root mean square error (RMSE) of the intervals, which cannot be directly converted to a mean relative or absolute error. As a result, both categories will present numerical analyses and comparisons that do not consider all of the reviewed works for that respective category. As a final remark, it is important to note that for all the works that were reviewed, none of them provided an explanation as to why the architectures and carriers were chosen in favor of other options, which means that the conclusions drawn in this work regarding these points do not factor in the reasons that these design choices were made in the first place.

### 4.1. Works Measuring Heart Rate

Out of all the works listed in the previous section, 56 works are analyzed in this category. Based on the criteria established in the same section, and since 25 of these works do not report their metrics in the form of the MAE, they will not be considered for the quantitative analysis that follows. They will, however, be considered when discussing the number of works for any given parameter.

#### 4.1.1. Hardware Architecture

For the works in this category, in terms of architecture usage, only one of the works did not specify which one they were implementing [56], and only one work reported using an architecture that differed from all the other ones that were mentioned [52]. As for the other architectures, the number of times that these are used is described in Figure 4.

As can be seen in the image, the FMCW architecture is used more times than both other architectures combined, while UWB is the one that is rarely used for this type of system. Based on the explanations provided in [10] regarding each of the architectures, certain points can be made. While CW radars are cheaper and easier to implement in terms of hardware, they are more heavily affected by forms of noise that do not affect these other two architectures because of their frequency modulation, such as multi-path noise. On the other side, UWB systems have much more complex hardware implementations and are harder to work with in terms of signal processing. FMCW radars strike a better balance with ease of use than UWB radars and have a more robust functionality than CW radars. With the recent increase in the availability of hardware platforms that allow for the implementation of more complex hardware platforms, FMCW radars have come to be the most researched and tested type of architecture in the field of radar-based heart rate monitoring.

As for the link between architecture and system performance, the boxplot displayed in Figure 5, which was made using the available data related to the MAE, was used to perform a comparisons and draw any possible conclusions.

As displayed in Figure 5, CW is the architecture with the higher median error when compared to both FMCW and UWB. It presents similar values to FMCW for maximum and minimum, but higher errors on average with regard to the other two. To compare FMCW and UWB, despite the significant difference in the number of works between the two architectures, it is still relevant to point out the similarities in their results. UWB shows a smaller interquartile range and a lower maximum value, therefore outperforming FMCW.

The results of this analysis point to a link between an improvement in system performance and the chosen system architecture, with FMCW being the better performing one when compared to CW. This is expected, given the fact that the architecture is inherently more robust to noise and signal interference due to how it functions. As for UWB, is is uncertain whether or not it definitively outperforms the other two architectures, and more research using UWB systems is needed to solidify this point.

#### 4.1.2. Carrier Frequency

For comparing the carrier frequencies that are used across the listed works in this category, they are separated into the following four frequency bands:Below 20 GHz;20 to 40 GHz;60 to 80 GHz;Above 120 GHz.

For the works in this category, only two did not specify the used carrier [32,33], and for the remaining works, the carriers used in each were split into the aforementioned bands based on their frequency for unmodulated systems and their corresponding center frequency for systems with frequency modulation. Their distribution across the bands is displayed in Figure 6.

As displayed in the image, carriers in the third band are the ones that are most used. This is directly related to the architecture usage trends, since most of the systems in this band use an FMCW radar, whereas for the first and second bands, the most predominant architecture is CW. All of the implemented UWB systems use carriers in the first band.

For the performance comparison of the carrier frequencies, a boxplot of the mean absolute error was made to analyze the distribution of errors across each individual bands, which is displayed in Figure 7. These results show that the second band, which is mostly composed of works using a carrier of 24 GHz, is the one that achieves the worst results, displaying not only the highest maximum error but also a higher median error, while the other two bands present very similar median values. The fourth band was not included in this analysis due to there only being two works using a system with a carrier of this band, which means it is not possible to accurately compare it to the others. As for works that performed HR measurements on different carriers, only [13] performed these tests, meaning that the connection between the carrier frequency and system performance is not widely researched for these systems. Despite the fact that it is not possible to definitively say whether the performance differences between the bands is due to the carriers or the architectures predominantly used in each one, since both the first and third bands present similar results, we can confirm that there is no inherent loss of performance associated with lower carrier frequencies; therefore, there is no inherent performance gain to be achieved by increasing them.

#### 4.1.3. Measurement Distance

Measurement distance is important for these systems because it affects the performance of the system in several different ways, the foremost of which being its effect on the power of the reflected signal $P_{r}$ in relation to the transmitted power $P_{t}$, as illustrated in the radar equation [10]:(1)$$P_{r}=\frac{P_{t}G^{2}\sigma \lambda^{2}}{{\left(4\pi \right)}^{3}R^{4}}=\frac{P_{t}G^{2}\sigma c^{2}}{{\left(4\pi \right)}^{3}f^{2}R^{4}}$$
where *G* is the antenna gain, λ is the wavelength, and σ represents RCS. This equation establishes an inverse relation between the distance *R* and $P_{r}$ and a similar relation between the carrier frequency and $P_{r}$. This relation is obviously extremely important because the more power the reflected signal has, the better the cardiac movement can be measured. The other effects that distance has are much less noticeable and apparent. Larger distances mean that there are more parts of the subject’s body that reflect the signal, which in turn increases its distortion. Inversely, placing the antennas too close to the subject’s body can also affect the reflected signal due to the differences between an antenna’s near field and far fields resulting in uncertain behaviour and distortion for the reflected signal.

An important part of the design of these systems is the environment in which they are meant to be used. With this in mind, beyond trying to quantify the impact distance has on system performance, it is also pertinent to analyze the conditions in which these systems are commonly tested to see how similar they are to the radar’s potential use cases. To that end, as displayed in Figure 8, the distances at which tests were performed across all works in this category were counted.

These distances lie mostly in the range below 1 m, with tests commonly being performed in increments of 0.5 m. As mentioned before, in an attempt to quantify the relation between distance and a performance metric, in this case the MAE, a linear regression was used to model the relation between distance and result error. The corresponding results are displayed in Figure 9, where there is a noticeable increase in the MAE as the distance also increases. Therefore, it is possible to link a variation in system performance based on the measurement distance, but no specific values could be determined.

### 4.2. Works Measuring Heart Rate Variability

Out of all the works listed in the previous section, 30 works are analyzed in this category. Based on the criteria established in the same section, since 16 of these works do not report their metrics in the form of the MRE, they will not be considered for the quantitative analysis that follows. They will, however, be considered when discussing the number of works for any given parameter.

#### 4.2.1. Hardware Architecture

The distribution of the architectures used across the works relevant to this category can be seen in Figure 10, where it is evident that both CW and FMCW see similar amounts of usage, whereas UWB is once again the least used type of system. As such, the UWB architecture cannot be accurately compared to the other two and will not be included in any comparison of results performed from this point forward. The remaining architectures still follow a similar usage pattern to the previous category, although CW systems are more predominant in this category. Since the authors never give a reason as to why a specific architecture is chosen, it is difficult to explain this difference in usage. One possibility is that, despite the fact that FMCW radars are more robust to noise and interference than CW, they are more expensive to develop, making them less worthwhile to use in research.

Comparing the results using what is presented in Figure 11, FMCW systems present better performance, as evidenced by the lower median error and smaller values for maximum and minimum error. This means that, on a general level, it is possible to argue that using the FMCW architecture will produce better results. This is explained by the aforementioned robustness of the architecture, which leads to a smaller dispersion of the results when compared to CW. These results follow a similar trend to the one displayed in Figure 5, which means that there is a clearly better performing architecture, which is FMCW. Lastly, since UWB is not as widely tested for these scenarios when compared to the other two architectures, it is not possible to say if (similar to the results for HR measurements) UWB would outperform the other two architectures. More tests are required to verify this possibility.

#### 4.2.2. Carrier Frequency

The comparison of the results based on the used carrier frequencies follows the same categorization into bands as mentioned in Section 4.1.2. The distribution of this category of work across these bands is displayed in Figure 12, where it can be seen that all bands are used in a more or less equal capacity.

Based on the boxplot displayed in Figure 13, which was made from the mean relative errors of the works in this category, it is possible to say that the third frequency band of 60 to 80 GHz leads to better results, presenting not only the lowest median of 1% but also low values for the extremes. The 0 to 20 GHz band presents a similar distribution but exhibits a higher median value of 2%. With regard to the second band specifically, it could simply be due to the fact that all of the systems using carriers in this band use the CW architecture, which, as mentioned before, is more susceptible to interference. In fact, the boxplot for this band and the boxplot for the CW architecture shown in Figure 11 are extremely similar. The same applies to the third band and the FMCW architecture. For the first band, the performance can be explained in very much the same way as the second one, where the predominant architecture used in this band is CW; however, it is also due to the fact that these frequencies do not have enough sensitivity to achieve the same kind of results that are achievable using carriers in the third band. Lastly, considering the fact that the reasoning for choosing a specific carrier is never given, and based on the analysis performed, it is possible to say that the frequency of the carrier signal does not have an impact on the performance of the system.

#### 4.2.3. Measurement Distance

As mentioned in Section 4.1.3, the distance at which measurements are taken is extremely important because it will affect the energy of the reflected signal, making it more or less susceptible to noise relative to the radar’s positioning. In the case of HRV measurements, this is important because there needs to be minimal loss of information regarding the cardiac activity in order to obtain reliable results. Despite this, the possible effects due to close proximity of the antennas to the subject still apply, so one cannot simply place them as close as possible, and, depending on the overall system and use case, the proper positioning needs to be determined.

Regarding the tests conducted for the works in this category, their distribution in terms of range is displayed in Figure 14 where, much like what was displayed in Figure 8, most of the tests are conducted at distances lower than 1 m. As for systems tested across different ranges, only four of the works in this category performed measurements in different ranges [13,70,76,79], following the trend of HR-focused systems where there are few tests being performed at ranges larger than 1 m.

Given the amount of works also presenting RMSE as an error metric, linear regression was also performed for this metric, alongside the one performed for MRE. The results of these regressions are presented in Figure 15, where it can be seen that there is an increase in the error values related to an increase in distance. This is effectively the same result as was achieved for the previous category of works.

### 4.3. Vital Sign Extraction Algorithms

Given the fact that HRV measurements rely on the measurement of the interval between consecutive signal peaks and HR measurement is performed either by measuring the number of peaks in a signal or by performing some form of frequency domain analysis, the algorithms used for signal processing will not be analyzed in separate categories. The algorithms used in each work were not listed in Section 3 for the sake of clarity. Lastly, all algorithms mentioned in this section are ones that have reported uses in radar-based systems, and it is in the context of these systems that they are analyzed.

The algorithms (not the full signal processing chain) used across the reviewed works can be split into several different classes based on their functionality (not all algorithms are listed):Digital Filtering, e.g., Bandpass, Moving Average;Spectral Analysis, e.g., Fourier Transform, Cossine Transform, CZT, FTPR;Mode Decomposition, e.g., EMD, EEMD, VMD, ICA;Wavelet Transform, e.g., DWT, MODWT, CWT, WPT;Deep Learning;Uncategorized, e.g., Autocorrelation, Differential Enhancement.

For almost all of the reviewed works, the signal processing chains use either zero-crossing detection or peak detection in order to detect the data points relevant for HR/HRV parameter calculation. However, some resort to more complex forms of event detection to avoid false positives and improve results such as the Viterbi Algorithm or HSMM [73,78] or the Decoding Peak Detection and Template Matching Algorithms [20,50,78]. As for the usage distribution of each of the aforementioned algorithm classes, it is difficult to tally them individually, since most of the time, they are used as parts of a larger signal processing chain, in which case there is no reason to count individual uses of the algorithm. Instead, it would make more sense to try and count the times a specific algorithm chain is used (for example, how many times mode decomposition is used in tandem with a wavelet transform), but that is also not reasonable because there are different pairings of mode decomposition and of the wavelet transform, which can lead to different results or even applications of the same algorithms that lead to different results due to different parametrization. This means that when comparing the results obtained in the works that were reviewed, the only outcome is a general idea of which algorithms might perform better in general. It is important to note that since there are works in which the details of the algorithms implemented are not disclosed, for example, exact data for the filters designed in [81] or which type of wavelet transform is used and the criteria used to choose the correct IMFs in [15,23], attempting to replicate the algorithms used in some of these works might lead to results that are different from what was expected.

Despite these constraints, it is still an important analysis to perform, since it allows us to obtain a general idea of which classes of algorithm are used more frequently, and how these classes are used together. In terms of performance in HR measurements, most algorithms can achieve good results (considering a minimum accuracy of over 95%), with mode decomposition, wavelets and digital filtering being the most used, either combined or individually. Although several types of the wavelet transform are used, none present a clear advantage when considering only the results. For mode decomposition, improvements on existing algorithms have been presented, but since no single improvement sees frequent use, it is hard to assert which one is better. FTPR also stands out with highly accurate results (≥99%), but as it is not frequently tested [43,44], it is uncertain whether this accuracy would be maintained with further testing. Signal processing methods using deep learning algorithms or other forms of neural network or machine learning are also used, albeit rarely in the reviewed works; however, they present good results. For HRV measurements, the algorithms that are used more frequently follow the trend for HR measurements, but works relying mostly on digital filtering and the simpler forms of mode decomposition tend to display higher error values. Signal processing chains that properly implement several algorithms in tandem [68,78] or improved/more complex versions of pre-existing algorithms [63,76] tend to be the ones that obtain the best results. Deep learning was also used in one work for HR measurements [60] and in one work for HRV measurements [65], with both achieving good results. Overall, there is a large overlap in terms of the algorithms used for measuring HR and HRV; however, systems measuring the latter tend to present more complex signal processing.

## 5. Discussion

The goal of this review is to assess, in terms of hardware architecture, carrier frequency and the signal processing chain, the choices of which tend to lead to the best possible performance based on the work that has already been performed for this kind of system. However, when comparing systems based on each of these categories individually, the impact of other categories cannot be separated. Because of this, the points made for each of the categories are not meant to be definitive, but more so guidelines to help researchers to make their decisions when designing their own systems.

With regard to the impact that the used architecture has on the performance of the system, if comparing based solely on the values for the median error metrics that were presented, then FMCW is the better choice for this kind of system. However, when considering other factors like cost and hardware complexity, then CW is still a viable alternative despite the fact that the architecture suffers from more interference than FMCW. With regard to the UWB architecture, given the performance displayed for HR measurements, it is still a viable choice presenting good performance. However, further testing and experiments for HRV measurements are required to truly determine how much better or worse it might perform when compared to the other two architectures. IRegarding the impact of the carrier frequency, systems with carriers in the range below 20 GHz and carriers above 60 GHz present better results for both categories. However in the context of this work, it is important to note that it is not possible to say for certain that the results presented are solely due to the used carrier frequency or if they are in part or solely due to the architectures that are used, since most systems using higher carrier frequencies tend to use FMCW, whereas systems with lower frequencies use mostly CW. Based on the relationship between the carrier frequency and the sensitivity to the chest wall displacement mentioned in Section 2.1, it was expected that systems with higher carrier frequencies would perform better than ones with lower frequencies. However, as can be seen in the results of the reviewed works, this is not the case. With this in mind, it is not possible to establish a definitive link between the used carrier frequency and the overall performance of the system. For the measurement distance, the decrease in performance due to larger distances was to be expected, not only due to the mathematical relation between distance and received signal power described by the radar equation but also due to the fact that larger distances mean that the signal will be reflected off of a larger surface of the subject’s body, which can cause significant amounts of interference. Apart from analyzing the extent of the impact on performance, analyzing the number of tests per distance also provided some insight into the current state of research on these systems. When analyzing this distribution, it became apparent that the number of experiments that were conducted at distances greater than 1 m is extremely low; for HR-focused works, out of 86 total tests, only 23 tests were performed at distances greater than 1 m. For HRV-focused works, out of 40 total tests, only 7 were performed at distances greater than 1 m. Since most of the reviewed works focus on studying the performance of specific signal processing methods, performing tests at distances that minimize interference and the impact of confounding variables is desirable. However, it is also important to perform tests in scenarios closer to the ones encountered in real-world applications. As such, an increase in testing performed in non-ideal scenarios is recommended. As for the signal processing performed, there is no singularly defined approach that is considered ideal. What is used is mostly dependent on what parameter the system is meant to be extracting: measuring HR only requires the preservation of the number of peaks within the measured signal, whilst HRV measurements also require the preservation of the position of the peaks with respect to time. This means that algorithms that measure HR through any form of spectral analysis, like the Fourier Transform, are not recommend for use in HRV measurements. Since there is not one individual signal processing chain/algorithm that is used frequently, it is not possible to determine which one would categorically lead to better results.

Lastly, in the closing of this review, based on the points made regarding the current state of the research on these systems, certain trends were seen that are worth trying to change. First, the standardization of the reported performance metrics is recommended in order to be able to properly judge different works on the same level. While certain works, for both HR- and HRV-focused systems, provide clear listings of the results per subject, there are still some that choose to report the metrics as an average of all subjects or use a different error metric. Some works report their performance solely in metrics that are not part of the norm, such as a correlation coefficient, or report their performance using only one type of metric. This makes a proper analysis of system performance significantly more difficult, since results can only be compared qualitatively (MRE and RMSE cannot be directly compared). Secondly, clearer explanations of the parameters necessary for some of the used signal processing algorithms are encouraged in order to improve replicability. This makes the testing of certain algorithms across different systems easier and opens up a path for research focused on comparing the most well performing algorithms and analyzing in which systems they perform better, if at all. Lastly, an increase in experiments conducted in non-ideal scenarios is recommended, in order to test the technology in conditions closer to the ones it will face if and when it becomes widely adopted.

As for recommendations for future work, although an assessment of which architecture/carrier frequency presents the best results is not really necessary, an assessment of which algorithm presents the best results by default for the widest variety of systems is a promising idea. Establishing a baseline system from which, if necessary, more specific implementations can be designed is a good step forward for a technology that is intended for use in the medical field and therefore needs to cover a variety of use cases.

## 6. Conclusions

Cardiac activity measurement using radar is a promising solution to the increased interest in the remote measurement of vital signs in smart-home environments and other scenarios in which contact is not possible. This review focuses on analyzing the performance of recent implementations of radar systems for cardiac activity measurement to try and link changes in performance to changes in several aspects of the overall system: hardware architecture, carrier frequency, and measurement distance. A general overview of the signal processing methods used in these systems was also carried out to establish possible usage trends. Based on the results of the performed analysis, it is possible to conclude that the hardware architecture and the signal processing methods used in the system are the aspects with the most significant impact, whereas the impact that the carrier frequency has is unclear. From this review, several recommendations are also given and mostly relate to the standardization of the reporting of results and the ease of replicating pre-existing systems. As more research is carried out on this topic, the focus should be on trying to find a baseline from which more specific systems can be designed.

## Figures and Tables

**Figure 1 sensors-24-07654-f001:**
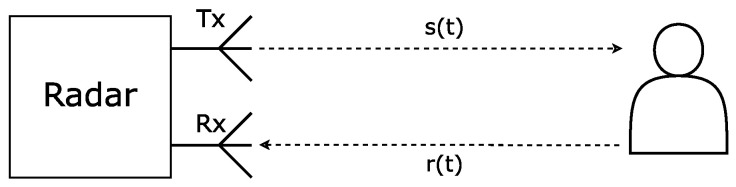
Basic radar architecture example.

**Figure 2 sensors-24-07654-f002:**
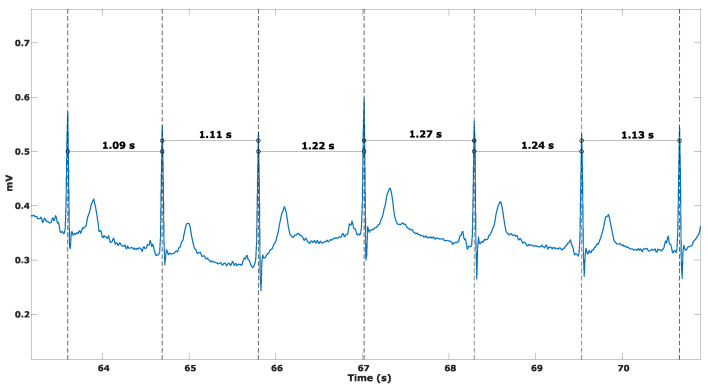
Beat-to-beat interval variation example.

**Figure 3 sensors-24-07654-f003:**
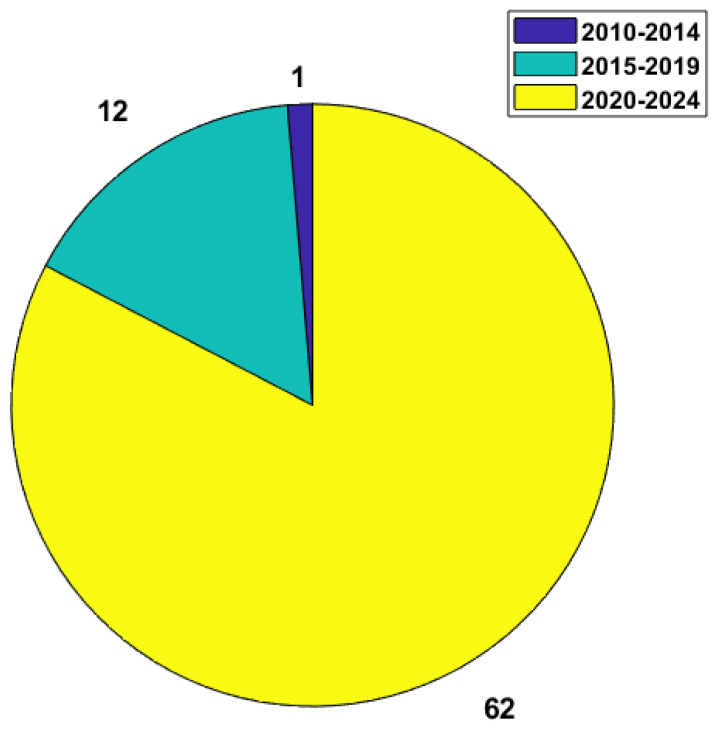
Number of works published per 5-year interval.

**Figure 4 sensors-24-07654-f004:**
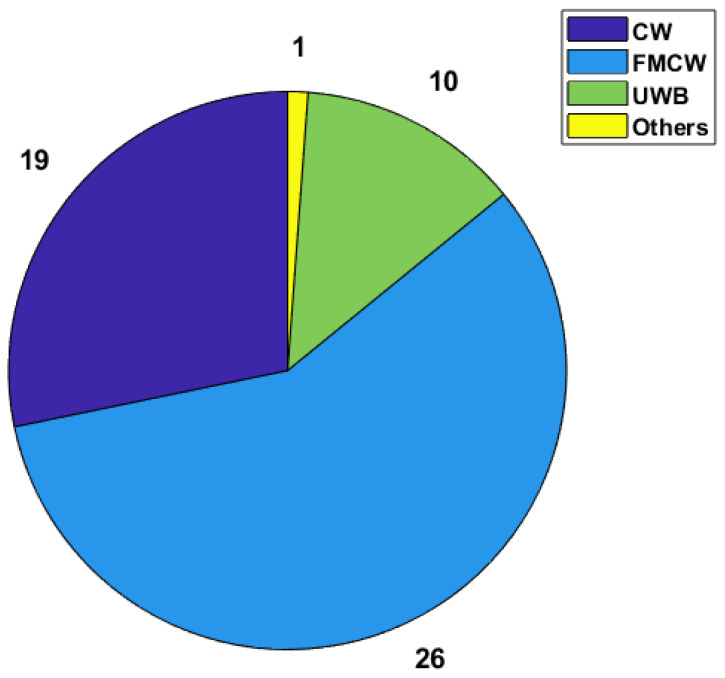
Architectures used in HR-focused works and the respective number of implementations.

**Figure 5 sensors-24-07654-f005:**
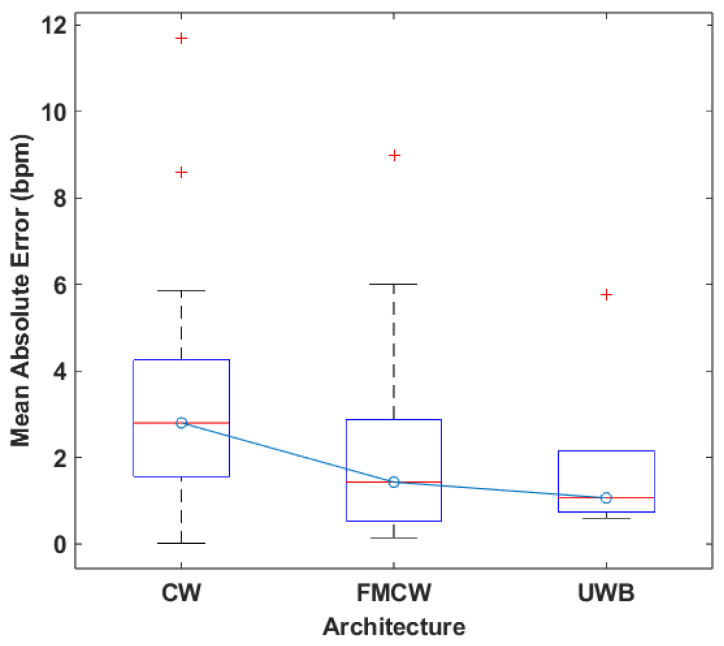
Boxplot of the MAE values for each architecture.

**Figure 6 sensors-24-07654-f006:**
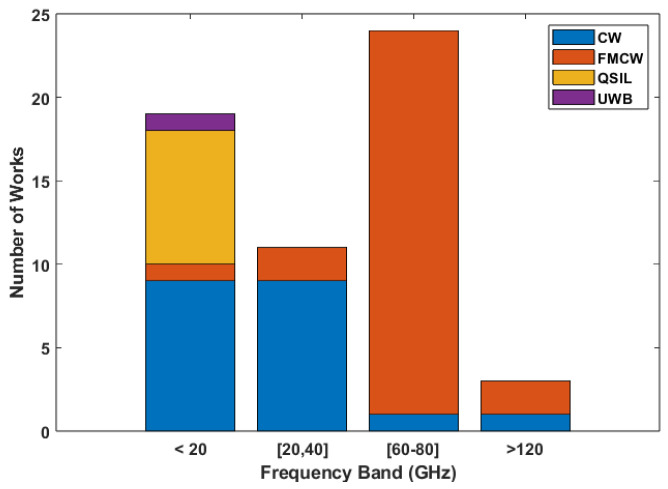
Number of works using carriers in each band based on their architecture in HR focused works.

**Figure 7 sensors-24-07654-f007:**
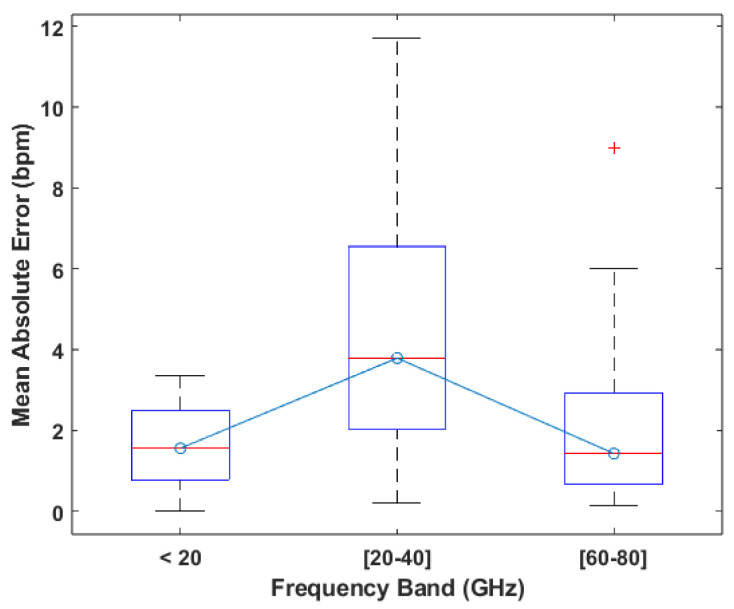
Boxplot of the MAE values reported per frequency band.

**Figure 8 sensors-24-07654-f008:**
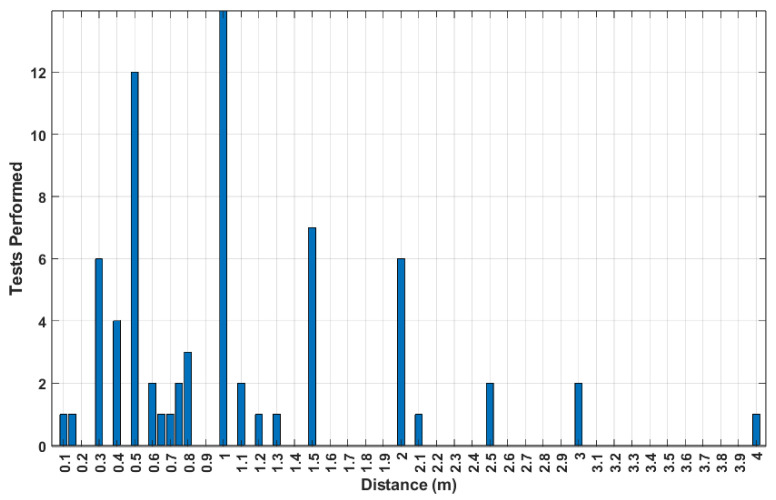
Number of HR-focused tests performed for each of the reported distances.

**Figure 9 sensors-24-07654-f009:**
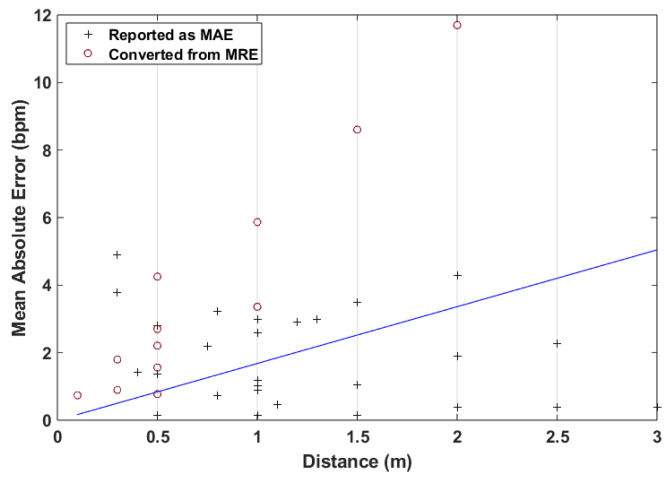
MAE distribution and linear regression with regards to distances tested in HR-focused works.

**Figure 10 sensors-24-07654-f010:**
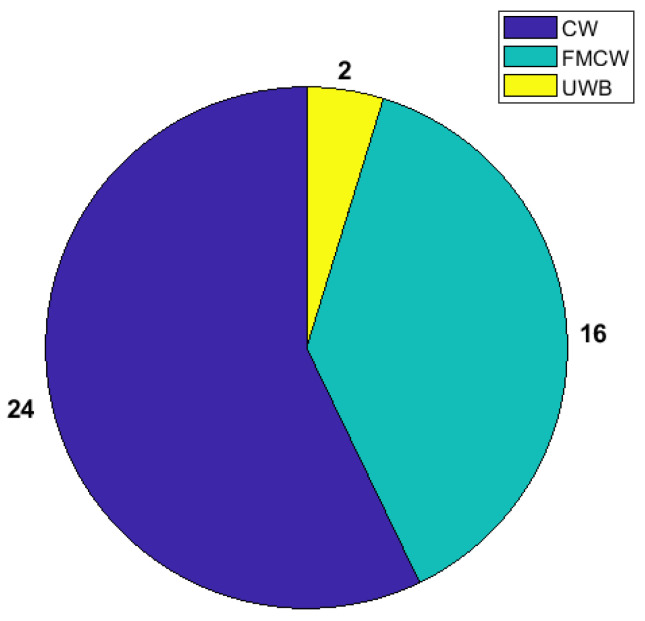
Architectures used in HRV-focused works and respective number of implementations.

**Figure 11 sensors-24-07654-f011:**
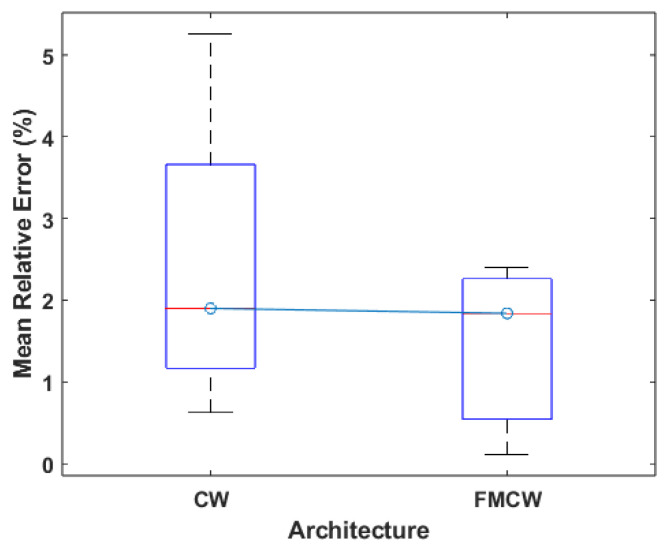
Boxplot of the MRE values for each architecture.

**Figure 12 sensors-24-07654-f012:**
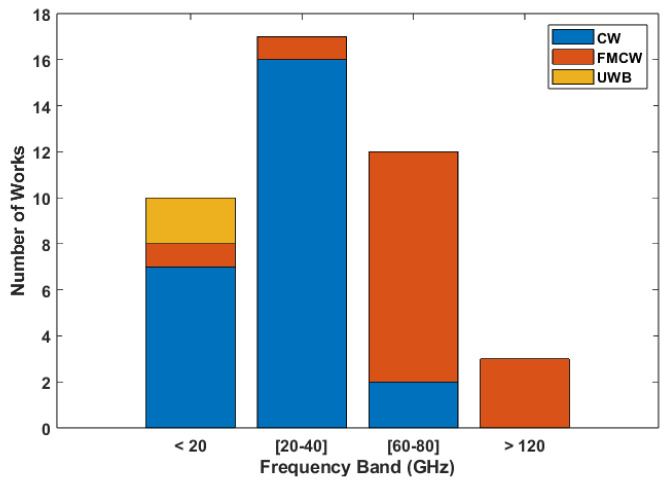
Number of works using carriers in each band based on their architecture in HRV focused works.

**Figure 13 sensors-24-07654-f013:**
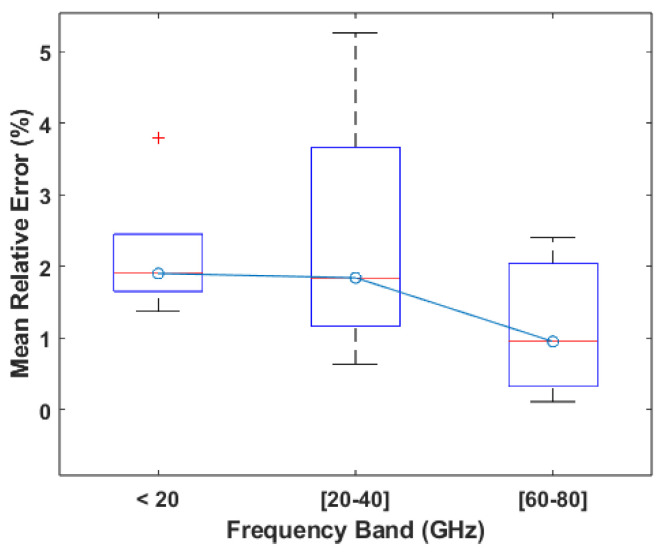
Boxplot of the MRE values reported per frequency band.

**Figure 14 sensors-24-07654-f014:**
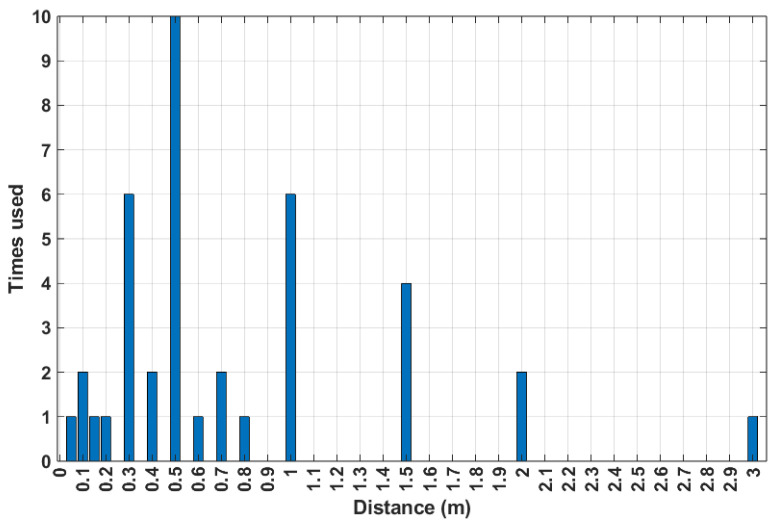
Number of HRV-focused tests performed for each of the reported distances.

**Figure 15 sensors-24-07654-f015:**
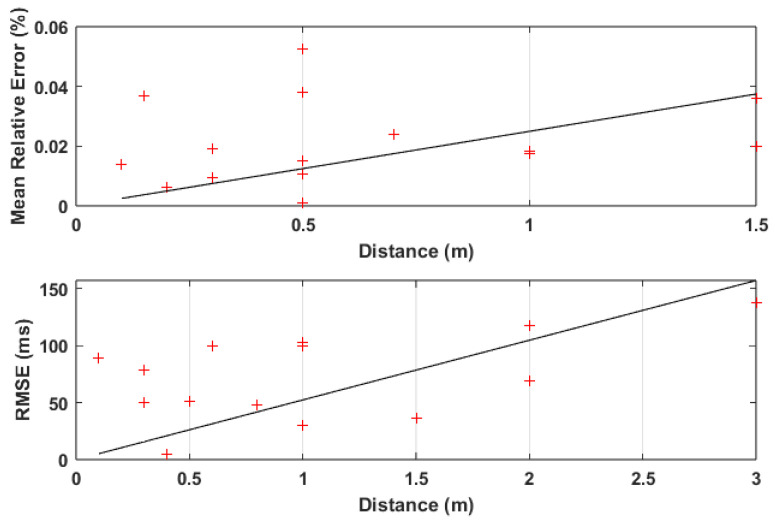
MRE/RMSE distribution and linear regression with regard to distances used in HRV tests.

**Table 1 sensors-24-07654-t001:** Listing of HR-focused works.

Reference/Pub. Year	Architecture	Carrier Freq. (GHz)	N° of Subjects	Distance (m)	HR Error Metric
[13]/2023	CW FMCW	24/134	1	0.3	MAE < 0.5 bpm
[14]/2023	FMCW	[60.25,64]	10	0.5	Accuracy = 98%
[15]/2021	CW	24	10	0.5	MRE = [7.09–19.5%]
[16]/2019	UWB	[2.9,10.1]	4	0.65; 0.8; 0.95	Accuracy = 89%
[17]/2022	CW	24	12	0.05	Correlation Coeff. = 0.96
[18]/2023	FMCW	77	7	0.5; 0.6; 1.2; 1.3	MAE = 2.79 bpm
[19]/2019	UWB	7.29	10	0.1	MRE = 1.23%
[20]/2023	FMCW	[77,81]	10	0.5	Accuracy > 98%
[21]/2023	FMCW	60	10	0.3; 0.5; 0.8; 1.1	MAE = 4.91 bpm
[22]/2023	CW	24	5	0.5	MRE 2.26%
[23]/2014	CW	5.8	10	0.5	MRE 3.68%
[24]/2023	UWB	[3.2,5.4]	10	–	Accuracy = 94.16%
[25]/2019	CW	2.45	6	0.4	Accuracy > 95%
[26]/2024	FMCW	[60,63.6]	13	–	MAE = 3.82 bpm
[27]/2018	CW	5.8	2	–	Accuracy = 96.5%
[28]/2019	CW	10.525	–	–	MAE = 0.014 bpm
[29]/2023	FMCW	60	8	0.5; 1; 1.5; 2; 2.5; 3	Accuracy = [85.71–97.48%]
[30]/2023a	FMCW	60	10	–	MRE = 2.924%
[31]/2023b	FMCW	[77,81]	7	1	MAE = 1.18 bpm
[32]/2024	UWB	–	30	–	MAE = 1.23 bpm
[33]/2023	IR-UWB	–	3	6	MRE = 9.96%
[34]/2023	FMCW	76.4	10	0.5	MAE = 0.15 bpm
[35]/2023a	FMCW	[77,81]	1	0.4	MAE = 1.43 bpm
[36]/2023b	CW	2.4	1	1.3	MAE < 3 bpm
[37]/2023c	FMCW	24	10	0.75	MAE = 2.2 bpm
[38]/2021	FMCW	[119.5,125.5]	10	1	Accuracy = 95.62%
[39]/2023	FMCW	[60,64]	3	[1.49–1.87]	MRE apporx 10%
[40]/2024	FMCW	[60,64]	6	[1.15–2.3]	MAE = 9 bpm
[41]/2020	LFMCW	[77,79]	2	1.1; 2.1	MRE = 2.56%
[42]/2024	FMCW	[77,79.6]	10	0.3; 1; 2	MRE = 1.4%
[43]/2018	CW	2.4	8	0.75; 1.5	Accuracy = 99%
[44]/2022	CW	77	2	0.3	MRE = 0.03%
[45]/2019	CW	2.45	1	0.3	MRE = 1.5%
[46]/2024	CW	24	7	0.6; 1	Accuracy = 95.25%
[47]/2023	FMCW	[77,81]	5	0.5; 1; 1.5; 2	MAE = 2.7 bpm
[48]/2023	IR-UWB	[5.9,10.3]	5	0.5; 1	Accuracy = [93–98]%
[49]/2022	CW	2.4	5	1	MAE = 2.6 bpm
[50]/2024	CW	2.4	4	–	MAE = 1.56 bpm
[51]/2022	CW	24	5	0.6	Poincare Comparison shows similarities
[52]/2023	QSIL	2.4	5	1	MRE = 5.6%
[53]/2023	FMCW	[58,64]	1	1	Accuracy = 74.4%
[54]/2024	FMCW	[77,81]	4	0.5; 1; 2; 3; 4	Accuracy = 100%
[55]/2022	CW	24	10	0.15	Correlation Coeff. = 0.998
[56]/2020	–	24	–	0.4	HR acc = 96%
[57]/2023	FMCW	[60,61]	2	0.3	MRE = 3.0%
[58]/2023a	UWB	7.3,1.4,23.328	13	1; 2	MAE = 0.9 bpm
[59]/2023c	UWB	[0.85,9.55]	5	0.5; 1; 1.5	MAE = 0.036 Hz
[60]/2023d	FMCW	[60,61]	3	–	Accuracy = 97.5%
[61]/2024	CW	24	30	0.4	PCC = 0.964
[62]/2020	FMCW	[8.15,8.65]	5	1.5	MRMSE ≈ 2 bpm
[63]/2019	CW	24	5	0.3	MAE = 3.79 bpm
[64]/2023	UWB	[6.5,8.1]	7	–	MRE < 1%
[65]/2024	FMCW	[77,81]	3	0.7	HR acc = 95.65%
[7]/2023	FMCW	[77,81]	10	1	MAE = 1.03 bpm
[66]/2023	IR-UWB	[6.765,9.04]	5	[0.5-5]	MRE = 86.5%
[67]/2023a	FMCW	[77,81]	5	0.5	MRE = 1.29%

**Table 2 sensors-24-07654-t002:** Listing of HRV-focused works.

Reference/Pub. Year	Architecture	Carrier Freq. (GHz)	N° of Subjects	Distance (m)	HR Error Metric	HRV Error Metric
[13]/2023	CW LFMCW	24/134	1	0.3;0.5	–	MAE < 10 ms
[15]/2021	CW	24	10	0.5;1;1.5;2	MRE = [7.09–19.5%]	MRE = 5.26%
[68]/2023	CW	24	2	0.4	–	RMSE = 5.2 ms
[17]/2022	CW	24	12	0.05	Correlation Coeff. = 0.96	ANN = 0.99
[69]/2024	CW	5.8	20	0.5	–	RMSE = 51 ms
[19]/2019	UWB	7.29	10	0.1	MRE = 1.23%	MRE = 1.38%
[20]/2023	FMCW	[77,81]	10	0.5	Accuracy > 98%	MRE = 0.11%
[22]/2023	CW	24	5	0.5	MRE = 2.26%	MRE = 11.48%/4.87%
[23]/2014	CW	5.8	10	0.5	Accuracy > 97%	BBI RE—2.53–4.83%
[25]/2019	CW	2.45	6	0.4	Accuracy > 95%	Accuracy = 96.78%
[70]/2022	FMCW	60	5	0.6	–	RMSE = 100 ms
[71]/2019	IR-UWB	8.748	1	0.7	–	High Correlation with ECG
[72]/2023c	FMCW	[23.8,24.8]	3	1	–	MRE = 1.84%
[38]/2021	FMCW	[119.5,125.5]	10	1	Accuracy = 95.62 %	MAE = 6.4 ms
[43]/2018	CW	2.4	8	1.5	Accuracy = 99%	MRE = 2%
[73]/2016	CW	24	5	1	–	Mean RMSE < 100 ms
[44]/2022	CW	77	2	0.3	MRE = 0.03%	MRE = 0.95%
[45]/2019	CW	2.45	1	0.3	MRE = 1.5%	MRE = 1.9%
[50]/2024	CW	2.4	4	–	MAE = 1.56 bpm	MAE = 9.31/12.42 ms
[74]/2023	CW	24	3	1.5	–	MRE = 3.61%
[75]/2022	CW	2.4	10	1	–	AAEP = 1.74%
[55]/2022	CW	24	10	0.15	Correlation Coeff. = 0.998	MRE = 3.68%
[76]/2021	FMCW	[76,81]	11	0.5	–	MAE = 3.89 ms
[77]/2024	CW	61	15	–	–	Accuracy = 93.8%
[78]/2021	CW	24	7	0.5	–	MRE = 1.15%
[62]/2020	FMCW	[8.15,8.65]	5	1.5	MRMSE ≈ 2 bpm	MAE < 5 ms
[79]/2018	CW	24	10	0.3	–	RMSE = 47.5 ms
[63]/2019	CW	24	5	0.3	AAEP < 5%	RMSE < 50 ms
[80]/2023	CW	24	18	0.2	–	MRE = 0.635%
[65]/2024	FMCW	[77,81]	3	0.7	Accuracy = 95.65%	AAEP = 0.86%

## Data Availability

All data relevant to this study is included in the article in Table 1 and Table 2. Further inquiries can be directed to the corresponding author.

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
