# Peer review of "Radar-Based Heart Cardiac Activity Measurements: A Review"

_sensors, 2024, doi:10.3390/s24237654_

Round 1

Reviewer 1 Report

Comments and Suggestions for Authors

This article provides a comprehensive review of non-contact physiological monitoring systems, aiming to analyze the impact of different systems on the performance of measuring heart rate (HR) or heart rate variability (HRV). While the overall structure and logic of the article are clear and rigorous, there are areas for improvement to enhance its academic contribution. 

1. the author's conclusion that the relationship between carrier frequency and performance is uncertain after comparing multiple research findings. It is suggested that author could analyze the relationship between carrier frequency and measurement performance by considering factors such as the millimeter-level displacement caused by heartbeats and the higher resolution of high-frequency signals.

2.  it is recommended that the author specify application scenarios, such as limiting the analysis of carrier frequency's impact on performance to situations where subjects are in a static state. This targeted scenario-based performance analysis may provide more actionable guidance for readers.

Reviewer 2 Report

Comments and Suggestions for Authors

This paper provides a comprehensive review of radar-based heart rate (HR) and heart rate variability (HRV) measurements. It is well organized and written. My comments are listed as follows.

In Figure 1, the total number of works is 75. In Figure 2, the total number of HR focused work is 85. The number of published works in two figures is not consistent.

Reviewer 3 Report

Comments and Suggestions for Authors

Based on works from relevant scientific databases, this review aims to evaluate the impact of various design choices on the performance of systems measuring Heart Rate or Heart Rate Variability. The paper is well-organized and presented, with meaningful contributions to the field. However, improvements are necessary before it can be considered for publication in Sensors, as follows:

1.     Present the paper's contributions in the Introduction section.

2.     Add a chart or figure in Section 2.1 to illustrate the radar architecture concept.

3.     Consider discussing ultrasonic radar in the paper. Several research works related to this subject can be found in literature, e.g., https://doi.org/10.3390/jsan8020032.

and https://doi.org/10.3390/diagnostics14192234.

4.     Provide an example to demonstrate the variability of heart rate signals in Section 2.2.

5.     Explain the significant increase in research on HR and HRV measurement systems from 2020-2024 compared to previous years in Figure 1.

6.     Add the publication year to the first column of Table 1 (e.g., [12]/2023), and so on.

7.     Merge Tables 1, 2, and 3 into a single table titled "Table 1: Listing of HRV-focused works,"

8.     Include some sensors used for heart rate measurement concerning vital signs in Section 4.3.

9.      Add a Conclusion section to the paper.

10.  Define all abbreviations at first mention, such as LFMCW.

11.  Provide a reference for Equation 1.

12.  Add grids to Figures 6 and 12.

Round 2

Reviewer 3 Report

Comments and Suggestions for Authors

The authors have addressed all my comments accordingly; I have no more suggestions. It can be accepted for publication in the current form.